# Composition and Antioxidative and Antibacterial Activities of the Essential Oil from *Farfugium japonicum*

**DOI:** 10.3390/molecules28062774

**Published:** 2023-03-19

**Authors:** Qiang Wei, Yi-Han Zhang

**Affiliations:** School of Medicine, Anhui Xinhua University, Hefei 230088, China

**Keywords:** *Farfugium japonicum*, chemical components, volatile oil, antioxidative activity, antibacterial activity

## Abstract

The composition of volatile oils of the leaf and stem of *Farfugium japonicum* (L.) Kitamura were prepared by supercritical fluid extraction (SFE)-CO_2_. A total 47 and 40 compounds were identified by GC/MS analysis, respectively, and only 13 compounds coexisted. The main constituent types in the leaf oil included alcohols (34.1%), hydrocarbons (24.1%), terpenoids (16.2%), benzenes (7.5%), and fatty acids (4.9%). In the stem oil, the constituent types chiefly included benzenes (18.8%), ketones (13.9%), terpenoids (17.0%), fatty acids (8.8%), phenolics (8.7%), steroids (8.6%), hydrocarbons (8.0%), and esters (5.7%). The predominant volatile compounds in the stem were 2-(1-cyclopent-1-enyl-1-methylethyl) cyclopentanone (11.7%), 1,2,3,4,5,6,7,8-octahydro- 9,10-dimethyl-anthracene (8.4%), 5-heptylresorcinol (6.5%), and *α*-sitosterol (5.2%). Those in the leaf mainly included (*E*)-3-hexen-1-ol (13.7%) and (*Z*)-3-hexen-1-ol (14.0%). This demonstrated a significant difference in the composition of both oils. Further study showed that stem oils demonstrated the highest DPPH (1,1-diphenyl-2-pinylhydrazyl) and ·OH free radical scavenging capacities at IC_50_ values of 9.22 and 0.90 mg/mL, respectively. In addition, they demonstrated the strongest antibacterial capacity against the Gram-positive bacteria methicillin-sensitive *Staphylococcus aureus* (MSSA) and methicillin-resistant *Staphylococcus aureus* (MRSA) at a minimum inhibitory concentration (MIC) of 0.16 mg/mL. This could be due to the SFE-CO_2_ extraction and the high accumulation of benzenes, terpenoids, and phenolics in the stem. In particular, the monoterpenes presented in terpenoids could play a special role in these findings.

## 1. Introduction

*Farfugium japonicum* (L.) Kitamura is a member of the Asteraceae family of herbal plants [1]. It has a wide distribution in East Asia, in countries such as Japan, Korea, and China. *F. japonicum* grows both in poor soils and in riverside areas, and it has a variety of leaf shapes, ranging from stenophyllous to spherical [2]. *F. japonicum* commonly exhibits large, heart-shaped leaves that are randomly spotted with yellow dots, and they look similar to a lotus leaves or a horseshoe.

In traditional Chinese medical therapy, *F. japonicum* rhizomes have been used to treat sore throats, colds, and coughs. In addition, modern pharmacological studies have shown its antipyretic, analgesic, antitumor, and anti-inflammatory activities [3]. It is reported that total flavonoids from the aerial parts of *F. japonicum* are more active against bacteria (*Bacillus subtilis*, *Salmonella*, *Staphylococcus aureus,* and *Escherichia coli*) than fungi (*Penicillium* and *Aspergillus* niger) [4]. Chemical compounds such as benzofuranosesquiterpenes (farfugin A and farfugin B) [5], furanoeremophilane (3β-angeloyloxy-10β-hydroxy-9β-senecioyloxyfuranoeremophilane and 3β-angeloyloxy-10β-hydroxyfuranoeremophilane) [6], and the pyrrolizidine alkaloid farfugine [7] have been isolated. Additionally, a previous report examining chemical composition of *F. japonicum* flower oil highlighted its main components and demonstrated anti-inflammatory effects [1].

Volatiles are secondary metabolites that are generated by many plants. They are also referred to as ethereal oils. They are described as being volatile, naturally occurring, and odorous compounds, and they are utilized in the production of various items, including food flavors, perfumes, cleaners, pesticides, herbicides, and medicines [8]. The active benefits of volatiles are demonstrated in a variety of ways, including the effects of excitement, antioxidation, bacteriostasis, and virustatic activities, demonstrating wide applications in terms of suitable medicine dosage forms, such as a tablet, lotion, spray, drop pill, soft capsule, microemulsion, gel paste, microcapsule, microsphere, liposome, polymer micelle, and so on [9].

The supercritical fluid extraction of the CO_2_ (SFE-CO_2_) process is becoming a highly appreciated, environmentally friendly, and effective plant extraction technology, and there have been increasingly broader applications over the past few decades. This avoids the shortcomings of traditional methods including poor purity, weak efficacy, solvent residue, and environmental pollution [10]. Compared to other fluids, supercritical CO_2_ is more suitable for the extraction of essential oils with its high solubility and diffusivity, safe and tasteless characteristics, and feasible critical temperature. To some extent, it restores the natural color, composition, and fragrance of the volatile oil with controllability [11].

According to previous studies [12,13], the terpenoids and phenols in volatiles are recognized for their antioxidant and antibacterial components. However, there are some important factors affecting the quality and ingredient composition in terpenoids and phenols, such as the extraction methods used, plants organs, and compounds ingredients etc. An SFE-CO_2_ extraction method may be one of the best choices for extracting volatile oils due to its advantages of CO_2_ compatibility with volatiles, high efficiency, greenness, and low temperature extraction. In terms of plant organs, the leaf and stem usually show differences in volatile compounds, leading to further diversity in their biological activities [14]. However, the mechanism producing such results has not been fully clarified. As an example, the volatiles involved in the structure of oxygenated terpenoids have been proven to have good antimicrobial and antioxidant capacities [15], and it is necessary to further test and verify the universal applicability of this to all plant organs. As a result, the current study aimed to extract the volatile oil via supercritical fluid extraction of CO_2_ (SFE-CO_2_). We present the analyses and comparison of the volatile oil composition from F. japonicum leaf and stem using GC/MS (gas chromatography/mass spectrometer) to compare the volatile compounds diversity, and further compare the antioxidative and antibacterial activities of the leaf oil and stem oil. We then discuss the difference between the leaf and stem volatiles from the perspective of biosynthesized procedure and relative enzymes and speculated regarding the possible constituents responsible for their antioxidative and antibacterial properties.

## 2. Results

### 2.1. Analysis of Essential Oil Compositions

The average content of leaf and stem volatiles were 1.1% and 0.6% *v*/*w*, respectively, on the basis of dry mass. Total ion chromatogram from leaf and stem oils in *F. japonicum* was seen in Figure 1. Table 1 shows that 73 constituents were successfully identified, with 47 compounds found in the leaf and 40 in the stem, accounting for 97.2% and 94.2% of the total volatiles, respectively. The dominant constituents of leaf oil were (*E*)-3-hexen-1-ol (13.7%), (*Z*)-3-hexen-1-ol (14.0%), tetratetracontane (4.7%), heneicosane (4.4%), and 1,2,3,4,5,6,7,8-octahydro-9,10-dimethyl-anthracene (4.2%), while those of the stem were 2-(1-cyclopent-1-enyl-1-methylethyl)cyclopentanone (11.7%), 1,2,3,4,5,6,7,8-octahydro-9,10-dimethyl-anthracene (8.4%), 5-heptylresorcinol (6.5%), *α*-sitosterol (5.2%), 1,3-dimethyl-benzene (4.4%), and 1,1’-bicyclohexyl (4.0%). This depicted a vast difference in leaf and stem compositions. Meanwhile, only 13 common compounds were presented in both the leaf and stem oil, accounting for 21.7% and 55.6% of the total volatiles, respectively. This demonstrated a significant difference between the composition of both oils.

Table 1 and Figure 2a showed a comparison of the number and relative amount between the leaf and stem volatiles compounds in *F. japonicum*, demonstrating that the types of chemical compound from the extracts of different plant parts could vary greatly. In total, 47 essential compounds were found in the leaf volatiles, containing 15 terpenoids, 7 hydrocarbons, 6 alcohols, 5 benzenes, 2 aldehydes, 2 esters, 2 phenolics, 2 fatty acids, 1 ketone, 1 steroid, and 1 coumarin. However, 40 essential compounds found in the leaf volatiles were identified including 9 hydrocarbons, 8 benzenes, 8 terpenoids, 3 esters, 3 phenolics, 3 fatty acids, 2 alcohols, 2 ketones, 2 steroids, and 1 aldehyde.

The relative amounts of different volatiles, as shown in Table 1 and Figure 2b, also revealed a significant difference between the leaf and stem. The compounds type in leaf oil mainly included alcohols (34.1%), hydrocarbons (24.1%), terpenoids (16.2%), benzenes (7.5%), and fatty acids (4.9%). In contrast, the stem oil was dominated by benzenes (18.8%), ketones (13.9%), terpenoids (17.0%), fatty acids (8.8%), phenolics (8.7%), steroids (8.6%), hydrocarbons (8.0%), and esters (5.7%). The compounds were mainly separated into five classes in leaf oil, while the stem contained eight classes. This could be due to significant constituent diversity in the stem due to different secondary metabolism processes. This might explain why the leaf and stem oils have significantly different flavors.

The main classification of *F. japonicum* flower volatile compounds found was hydrocarbons in a previous study, including caryophyllene, 1-undecene, and 1-nonene [1]. This differed from the volatile compounds found in the *F. japonicum* leaf and stem in the present study. However, two compounds were identified in both studies: caryophyllene oxide, which coexisted in the leaf and flower oil at total ratios of 1.9% and 0.9%, respectively, and diethyl phthalate, which coexisted in the leaf, stem, and flower oil at total ratios of 1.4%, 2.7%, and 1.5%, respectively. The findings also revealed that the flower oil composition of volatiles differed significantly from that of the leaf or stem.

The generation of plant volatiles is highly organized based on genetic and ecological diversity at different development stages [16]. As seen in Figure 3, two pathways of producing methyl-D-erythritol phosphate (MEP,) and mevalonic acid (MVA) exist in terpenoid biosynthesis and regulation. This includes a process of converting small plant molecules to complex and diversified structures via related enzymes, including the creation of general precursors, isopentenyl diphosphate (IPP), the double-bonded isomer dimethylallyl diphosphate (DMAPP), the direct precursors geraniol diphosphate (GPP) and farnesyl diphosphate (FPP), and terpenoids [17]. All terpenoids in nature have IPP and DMAPP as their basic synthesis units, further producing direct precursors of GPP, FPP, and geraniol diphosphate under the action of enzymes. GPP and FPP represent the natural shared stage of terpenoids synthesized by plants. Geraniol diphosphate, at the most important stage of terpenoid synthesis, involves a greater amount of terpene synthases and modifying enzymes, and is directly related to varied and complex terpenoids [18,19,20].

Enzymes involved in terpenoid biosynthesis that encoded genes expression and regulation are key research domains. There are three main rate-limiting enzymes in the MVA pathway: 3-hydroxy-3-methylglutaryl CoA reductase (HMGR), 5-phosphomevalonate kinase (PMK), and mevalonate kinase (MK). They provide a critical regulatory site in the terpene biosynthesis pathway in the cytoplasm [22]. HMGR is encoded by the gene CcPMK [23]. As an example, the high content of terpenoids in Cinnamomum camphora is correlated with the high expression levels of the HMGR and PMK genes, as the upstream terpenoid synthesis pathway. MK triggers the transfer of phosphate groups to produce mevalonate-5-phosphate (MVAP) [24]. The IPP/DMAPP synthase-related gene TmHDR has shown expression in all plant tissues [25].

As for the MEP pathway, 1-deoxy-D-xylulose-5-phosphate synthase (DXS) and 1-deoxy-D-xylulose-5-phosphate reduction of the isomeras (DXR) are the major rate-limiting enzymes [21].

In this work, analysis of leaf and stem oils demonstrated a similar total terpenoid content, accounting for 16.2% and 17.0%, respectively. However, the monoterpenes-rich stem volatiles may have resulted from the highly expressed genes of the rate-limiting enzymes HMGR, PMK, and MK in the MVA pathway, or DXS and DXR in the MEP pathway. This could directly relate to their stronger antioxidative and antibacterial capacities, but more research is needed to verify this speculation in *F. japonicum*.

### 2.2. Antioxidative Capacity

Table 2 shows the IC_50_ values for both the leaf and stem essential oils from DPPH (1,1-diphenyl-2-pinylhydrazyl) and ·OH assays. Essential oils from the *F. japonicum* stem showed the strongest DPPH and ·OH free radical scavenging capacities, exhibiting IC_50_ values of 9.22 and 0.90 mg/mL, respectively. This likely correlates with the high content of benzenes, phenolics, and terpenoids. According to a previous study, the antioxidative capacities were dependent on the side chain structures and substitution patterns on the benzene ring [26]. Monoterpenes and diterpenes usually produced the effect of quenching of singlet oxygen, hydrogen transfer, or electron transfer, which led to their antioxidant activity [27]. Meanwhile, high phenolic content was correlated with antioxidant potential, which was highly redox active and exhibited a crucial function in free radical neutralization and peroxide breakdown [28,29]. Leaf volatiles were abundant, as shown in Table 1 and Figure 1, with 15 terpenoids and 5 benzenes accounting for 16.2% and 7.5%, respectively. However, the stem volatiles were present in eight benzenes, eight terpenoids, and three phenolics, accounting for 18.8%, 17.0%, and 8.7% of the total contents, respectively. In particular, the monoterpenes of stem volatiles, such as β-linalool (2.9%), 4-terpeneol (1.7%), α-terpineol (1.7%), trans-nerolidol (3.2%), boronia butenal (2.2%), and 3,5,6,7,8,8α-hexahydro-4,8α-dimethyl-6-(1-methylethenyl)-2(1H)nahpthalenone (2.7%), had a significant accumulated advantage associated with the presence of various active stem volatiles, providing important support for a higher antioxidative capacity [30].

Additionally, the antioxidative capacity of volatiles was considerably impacted by time and pressure based on a DPPH experiment [31]. According to the findings, SFE performed under high pressure and for a lengthy period can provide lavender oil with a significant level of antioxidative capacity, while a large yield of volatiles may also be obtained via extraction at lower temperatures, pressures, and times (36.6 °C, 10 MPa, and 73.6 min, respectively). However, the DPPH scavenging capacity was subsequently reduced. Consequently, a pressure of 50 MPa and a time of 120 min could be conducive to the antioxidation process due to the obvious abilities of scavenging DPPH and ·OH radicals.

### 2.3. Antibcterial Capacity

Table 3 presented antibacterial capacities using the minimum inhibitory concentrations (MICs) of leaf and stem oils, which demonstrated that all bacteria were more sensitive to stem volatiles than leaf volatiles. In particular, the stem oil showed strong activity against Gram-positive (G^+^) microbes, such as methicillin-sensitive *Staphylococcus aureus (MSSA)* and methicillin-resistant *Staphylococcus aureus (MRSA)* (MIC = 0.16 mg/mL), while exhibiting weak capacity against Gram-negative (G^−^) microbes, including *Pseudomonas aeruginosa*, *Escherichia coli*, *Proteus* spp., and *Klebsiella pneumoniae*.

As antibiotic ineffectiveness due to drug resistance has been a severe clinical problem, particularly in relation to hospital-acquired infections and animal-derived antibiotic diseases, plant-based antibacterial compounds are arousing wider interest. One of the intrinsic qualities of volatile oils is their antimicrobial activity; β-linalool, one of the stem volatiles, is reported to have antibacterial characteristics as a preservative against pathogenic germs [32].

According to several additional studies on trans-cinnamaldehyde, eugenol, or citral, the antibacterial action of volatiles showed both single and multiple target capacities. The hydrophobicity of volatiles made it significantly easier for them to travel through the lipid layer of bacterial cell membranes. In particular, they destroyed the function and structural components of the cell membrane, and they inhibited enzymatic processes and further disrupted the architecture of the cell walls to make them more permeable. Under certain circumstances, volatiles changed the membrane’s permeability by a different mechanism, namely by destroying the electronic transport, which triggered a rise in the cellular adenosine triphosphate (ATP) content. After the electron transport system was inhibited, it triggered a rise in energy, proteins, and other cellular components; furthermore, it disturbed the force that promotes the movement of protons across membranes, lowered the electrochemical potential, and finally led to cell lysis and death [25,33]. In addition, cell death also resulted from the release of ions and other cellular components caused by this shift in membrane permeability [34]. However, antibacterial capacities were strongly correlated with volatile components, with phenols showing obvious inhibition, ketones showing weaker activity, and hydrocarbons being almost inactive [25].

Terpenoids are important components found in volatiles, and they have methyl groups or oxygen atoms that combined with specific bacterial enzymes or are directly localized to bacteria. As an example, β-linalool has improved antibacterial properties because of the existence of powerful effective groups that delocalize electrons [35].

As previously discussed, stem volatiles had stronger antibacterial capacities due to their eight benzenes, eight terpenoids, and three phenolics constituents with total ratios of 18.8%, 17.0%, and 8.7%, respectively. The monoterpenes of the stem volatiles played a key role, including β-linalool (2.9%), 4-terpeneol (1.7%), α-terpineol (1.7%), trans-nerolidol (3.2%), boronia butenal (2.2%), etc. Therefore, the results demonstrated the monoterpenes had a vital impact on the antioxidative and antibacterial activities of volatiles, showing some difference with the reported oxygenated terpenoids [14]. This is perhaps due to almost all terpenoids in the leaf and stem being oxygenated terpenoids, except for compounds **27** and **36**, accounting for the similar content of terpenoids, at 15.5% and 14.4%, respectively.

Moreover, as is shown in Table 3, the stem volatiles showed a stronger inhibition of G^+^ microbes (MSSA and MRSA) at a MIC of 0.16 mg/mL, compared with G^−^ microbes. Due to the existence of a peptidoglycan layer in the bacterial outer membrane, G^+^ microbes often have increased sensitivity to volatiles. A lipopolysaccharide-joined double layer of phospholipids attached to the inner membrane (LPS) forms the outer membrane structure in G^−^ microbes. The presence of lipid A and O-side polysaccharide chains has been proven to be relevant to the resistance of G^-^ microbes to volatiles [36].

## 3. Materials and Methods

### 3.1. Plant Material and Reagents

The leaf and stem of *F. japonicum* (1000 g of each) were harvested separately from the campus yard of Anhui Xinhua University in China. The specimens of the leaf and stem (No.: AHXH 160 and 161) were identified by Prof. Qizhao Li and stored in a specimen room. The plant was shaded at an ambient temperature, cleaned, and ground to a fine powder.

### 3.2. Essential Oils Extraction

Both plant organs, the leaf and stem, each weighing 260 g, were put into an SFE-CO_2_ cartridge under the extraction process parameters of 50 MPa pressure, 40 °C temperature, and 120 min. The volatile oil in the extract collected from the outlet valve of the separation kettle was then isolated by hydrodistilation and was stored at 4 °C.

### 3.3. GC/MS Analysis

On an Agilent 5975C GC/MS (Agilent Technologies, Santa Clara CA, USA) with a HP-5MS column (5% phenyl methyl siloxane, 30 m × 0.25 mm, 0.25 μm), the volatiles were subjected to GC analysis. At a flow rate of 1 mL/min, helium was utilized as the carrier gas. After being set at 45 °C for 1 min, the temperature was elevated to 250 °C at 10 °C/min, maintained at 250 °C for 50 min, and then programmed to hold at 280 °C for 1 min. A split ratio of 40:1 was used with a 1 μL sample injection in split mode. The quadrupole was heated to 220 °C, and the ion source temperature was adjusted to 280 °C. The multi-channel plate voltage of 70 eV was applied, and the detector was set to operate in EI mode with a *m*/*z* range of 50–500.

All compounds identification was performed using the mass spectra, linear retention index (LRI), and Kováts retention index (KRI) data. The retention indices (RI) of every compound were calibrated with C5–C30 alkanes, comparing them to the mass spectra in the NIST libraries. By using peak area normalization [37,38,39], the relative amounts of the identified volatile compounds were quantified.

### 3.4. Antioxidative Activity Assay

#### 3.4.1. DPPH Inhibition Test

The antioxidative effect of essential oils was evaluated with DPPH inhibition activity as a reagent, with some modifications to the Kirby and Schmidt method [40]. A total of 50 mL of the sample solutions was blended with 500 mL of an alcoholic DPPH solution containing 4% (*w*/*v*) dichloromethane after the samples were dissolved in dichloromethane at 0.01–10 mg/mL. The mixture was left in the dark for 20 min at room temperature. Reading the absorbance at 517 nm allowed for the measurement of the DPPH radical’s inhibition. Equation (1) was used to determine the percentage of inhibition: (1)$$\mathrm{Inhibition}=\left[\frac{\left(A_{0}-A_{i}\right)}{A_{0}}\right]\times 100$$
where *A*_0_ represents the absorbance of the blank and *A_i_* represents the absorbance of the samples. A non-linear regression approach was used to calculate the IC_50_ values. All tests were performed in triplicate. A positive control of vitamin C was employed.

#### 3.4.2. Hydroxyl Radical (OH) Inhibition Test

Hydroxyl free radical inhibition of the essential oils was measured according to the procedures reported in the literature [41] with some changes. A total of 3 mL of iron sulfate (9 mmol/L), 3 mL of 2-hydroxybenzoic acid solution with 95% ethanol (9 mmol/L), and 1 mL of successive concentrations of CCBLP solution (0, 0.5, 1.0, 1.5, 2.0, 2.5, 3.0 mg/mL) were added to a 25 mL volumetric flask. Then, the solution was mixed with 3 mL of H_2_O_2_ (8.8 mmol/L) to keep the reaction going for 30 min at 37 °C. It was then diluted to 25 mL with distilled water. The absorbency was determined at 510 nm against a blank solution, and the oils and vitamin C were used as contrast agents. The equation in Section 3.4.1 was used to calculate the percentage inhibition.

#### 3.4.3. Antimicrobial Capacities Assay

The disc diffusion test was used to assess the volatiles of *F. japonicum* in terms of their antimicrobial properties [42]. The following strains were collected from the Anhui Provincial Center for Disease Control and Prevention of China: *MSSA*, *MRSA*, *P. aeruginosa*, *E. coli*, *P.* spp., and *K. pneumoniae*. A Muller–Hinton agar medium was suspended in 1.0% saline to create the bacterium inocula. A total of 5.00, 2.50, 1.25, 0.62, 0.31, and 0.16 mg/mL were obtained after 10% *v*/*v* dimethylsulfoxide (DMSO)-soluble essential oils were diluted at a concentration of 10 μg/mL. Blank discs containing 20 μL of 10% DMSO were used as the controls. The MIC was determined after 24 h of incubation at 30–35 °C in microtiter plates.

## 4. Conclusions

The chemical, antioxidative, and antibacterial properties of leaf and stem oils were studied. The GC/MS method revealed 73 different constituents, 47 of which were found in the leaf and 40 were found in the stem; of these, only 13 common compounds were found in both the leaf and stem oils, with a content of 21.7% and 55.6%, respectively. The main constituents in the leaf oil included alcohols (34.1%), hydrocarbons (24.1%), terpenoids (16.2%), benzenes (7.5%), and fatty acids (4.9%). In the stem oil, the constituents were chiefly benzenes (18.8%), ketones (13.9%), terpenoids (17.0%), fatty acids (8.8%), phenolics (8.7%), steroids (8.6%), hydrocarbons (8.0%), and esters (5.7%). The representative compounds in the leaf oil were (*E*)-3-hexen-1-ol (13.7%), (*Z*)-3-hexen-1-ol (14.0%), tetratetracontane (4.7%), heneicosane (4.4%), and 1,2,3,4,5,6,7,8-octahydro-9,10-dimethyl-anthracene (4.2%), whereas the stem oil mainly contained 2-(1-cyclopent-1-enyl-1-methylethyl)cyclopentanone (11.7%), 1,2,3,4,5,6,7,8-octahydro- 9,10-dimethyl-anthracene (8.4%), 5-heptylresorcinol (6.5%), *α*-sitosterol (5.2%), 1,3-dimethyl-benzene (4.4%), and 1,1’-bicyclohexyl (4.0%). This depicted a vast difference in the leaf and stem compositions.

Additionally, the stem oil presented apparent antioxidant and antibacterial capacities. This was likely linked to its high accumulation of benzenes, phenolics, and terpenoids, especially monoterpenes. Monoterpenes, as representative constituents with antioxidative and antibacterial capacities, demonstrated a different distribution or accumulation in the leaf and stem volatiles. This was possibly based on the stem advantages in terms of regulating the enzymes responsible for generating volatiles, namely, HMGR, PMK, and MK in the MVA pathway or enzymes DXS and DXR in the MEP pathway as well as their relative highly expressed genes. The findings also exhibited strong inhibition for G^+^ microbes (MSSA and MRSA) as well as weak inhibition for G^−^ microbes (*P. aeruginosa*, *E. coli*, *P.* spp., and *K. pneumoniae*). This could be related to the inhibition properties of volatiles and may be worthy of further study and exploitation.

In this work, constituent type, chemical compounds, relative content, and antioxidant and antibacterial capacities in *F. japonicum* leaf and stem oils were studied. This revealed a large difference in both plant organs. SFE-CO_2_ extraction revealed a high accumulation of monoterpenes and this could play a special role in these findings. This may provide could provide a direction for more extensive research in the future.

## Figures and Tables

**Figure 1 molecules-28-02774-f001:**
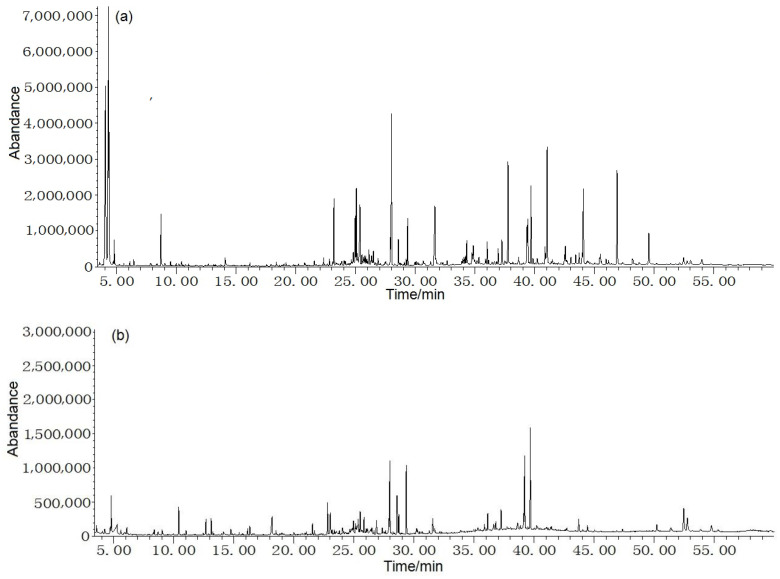
Total ion chromatogram from leaf (**a**) and stem (**b**) oils in *F. japonicum*.

**Figure 2 molecules-28-02774-f002:**
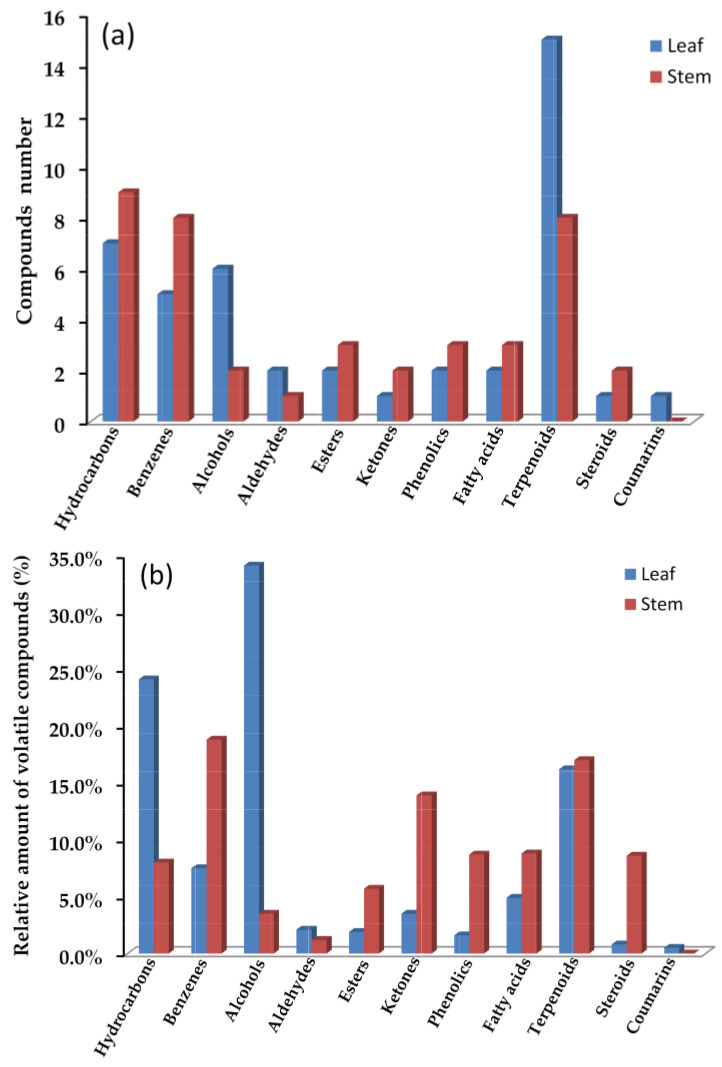
Comparison of volatile compounds from leaf and stem oils in *F. japonicum*: compounds number (**a**) and relative amounts of volatile compounds (%) (**b**).

**Figure 3 molecules-28-02774-f003:**
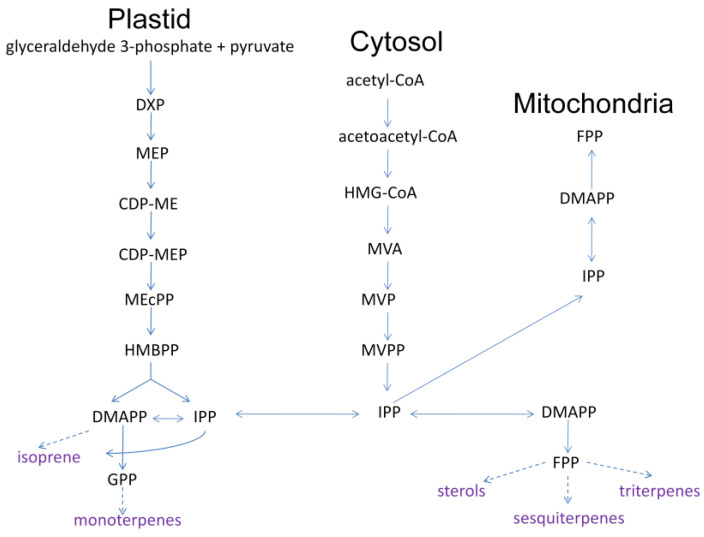
MEP and MVA pathway of terpenoids biosynthesis in plants (DXP, deoxyxylulose-5-phosphate; CDP-ME, 4-CDP-2-C-methyl-D-erythritol; CDP-MEP, CDP-ME 2-phosphate; MEcPP, 3-methyl-1,2,3,4-tetrahydroxybutane-1,3-cyclic bisphosphate; HMBPP, (E)-4-Hydroxy-3-methylbut-2-enyl diphosphate; HMG-CoA, 3-hydroxy-3-methylglutaryl CoA; MVP, 5-phosphomevalonate; MVPP, 5-diphosphomevalonate) (Adapted with permission from Ref. [21]. 2022, Du, Y.; Zhou, H.).

**Table 1 molecules-28-02774-t001:** Volatile compounds of leaf and stem from *Farfugium japonicum*.

No.	Compounds	LRI	KRI	RA/%	
				Leaf	Stem
1	Furfural ^(Ad)^	812	810	-	1.2
2	*cis*-1,2-Dimethyl-cyclohexane ^(H)^	822	820	-	0.6
3	Ethyl-cyclohexane ^(H)^	827	829	-	2.3
4	(*E*)-3-Hexen-1-ol ^(Ac)^	849	847	13.7	-
5	(*Z*)-3-Hexen-1-ol ^(Ac)^	851	847	14.0	-
6	1,3-Dimethyl-benzene ^(B)^	852	853	-	4.4
7	Benzaldehyde ^(Ad)^	964	964	0.2	-
8	Hydroperoxide, hexyl ^(Ac)^	977	978	3.5	-
9	1,2,4-Trimethyl-benzene ^(B)^	986	989	-	0.1
10	Trimethylenenorbornane ^(H)^	990	990	-	0.1
11	Decane ^(H)^	1002	1000	-	0.1
12	1,2,3-Trimethyl-benzene ^(B)^	1005	1005	-	0.4
13	*β*-Linalool ^(T)^	1098	1098	-	2.9
14	1,2,4,5-Tetramethyl-benzene ^(B)^	1108	1109	0.5	0.9
15	2-Ethyl-1-hexanol ^(Ac)^	1010	1010	1.7	3.4
16	1,4-Diethyl-benzene ^(B)^	1045	1046	1.2	3.6
17	Benzeneacetaldehyde ^(Ad)^	1047	1048	1.9	-
18	1-Ethyl-2,3-dimethyl-benzene ^(B)^	1093	1096	-	0.9
19	4-Ethenyl-1,2-dimethyl-benzene ^(B)^	1098	1100	-	0.1
20	Pentyl-cyclohexane ^(H)^	1123	1121	-	0.1
21	4-Terpeneol ^(T)^	1160	1160	-	1.7
22	*α*-Terpineol ^(T)^	1170	1172	-	1.7
23	2,3-Dihydro-benzofuran ^(B)^	1185	1188	0.5	-
24	Naphthalene ^(B)^	1191	1190	1.1	-
25	Dodecane ^(H)^	1201	1200	-	0.6
26	2-Methoxy-4-vinylphenol ^(P)^	1272	1272	-	1.0
27	Azulene ^(T)^	1296	1296	-	2.6
28	1,1’-Bicyclohexyl ^(H)^	1302	1304	1.9	4.0
29	5,5,8*α*-Trimethyl-3,5,6,7,8,8*α*-hexahydro-2H-chromene ^(T)^	1308	1309	0.7	-
30	n-Decanoic acid ^(F)^	1360	1360	-	3.6
31	2,6,8-Trimethylbicyclo [4.2.0]oct-2-ene-1,8-diol ^(T)^	1371	1370	0.4	-
32	Tetradecane ^(H)^	1402	1400	-	0.1
33	Curzerene ^(T)^	1480	1480	1.2	-
34	5-Isopropylidene-6-methyldeca-3,6,9-trien-2-one ^(K)^	1492	1494	-	2.2
35	2-(1-Cyclopent-1-enyl-1-methylethyl)cyclopentanone ^(K)^	1495	1497	3.5	11.7
36	*cis*-Calamenene ^(T)^	1510	1511	0.7	-
37	Shyobunone ^(T)^	1518	1518	2.3	-
38	2,4-bis(1,1-Dimethylethyl)-phenol ^(P)^	1539	1540	0.4	1.2
39	*trans*-Nerolidol ^(T)^	1547	1548	0.2	3.2
40	2,6,10-Trimethyl-tetradecane ^(H)^	1555	1557	0.6	0.1
41	n-Dodecanoic acid ^(F)^	1560	1561	-	3.2
42	2,6-Dimethyl-10-methylene-12-oxatricyclo [7.3.1.0(1,6)] tridec-2-ene ^(T)^	1577	1576	2.3	-
43	Caryophyllene oxide ^(T)^	1580	1581	1.9	-
44	Boronia butenal ^(T)^	1582	1584	-	2.2
45	Calarene epoxide ^(T)^	1592	1592	0.2	-
46	Diethyl phthalate ^(E)^	1594	1594	1.4	2.7
47	Ledene oxide-(Ⅱ) ^(T)^	1631	1631	0.7	-
48	*α*-Cyperone ^(T)^	1672	1673	0.4	-
49	6-Isopropenyl-4,8*α*-dimethyl-1,2,3,5,6,7,8,8*α*-octahydro-naphthalen-2-ol ^(T)^	1714	1714	0.7	-
50	3,5,6,7,8,8*α*-Hexahydro-4,8*α*-dimethyl-6-(1-methylethenyl)-2(1H)nahpthalenone ^(T)^	1790	1790	0.6	2.7
51	5-Heptylresorcinol ^(P)^	1830	1831	1.2	6.5
52	*trans*-9-Hexadecen-1-ol ^(Ac)^	1866	1868	-	0.1
53	1,1,4,6-Tetramethyl-perhydrocyclopropa [e] azulene-4,5,6-triol ^(T)^	1867	1869	0.5	-
54	1,2,3,4,5,6,7,8-Octahydro-9,10-dimethyl-anthracene ^(B)^	1878	1879	4.2	8.4
55	2-Methyl-9-(prop-1-en-3-ol-2-yl)-bicyclo [4.4.0] dec-2-ene-4-ol ^(T)^	1902	1904	1.7	-
56	*bi*-1-Cycloocten-1-yl ^(H)^	1941	1942	3.1	-
57	n-Hexadecanoic acid ^(F)^	1963	1963	3.5	2.0
58	Eicosane ^(H)^	2000	2000	3.5	-
59	1-Hexadecanol, acetate ^(E)^	2008	2009	0.5	-
60	Oralic acid, cyclohexyl octyl ester ^(E)^	2010	2010	-	1.0
61	Heneicosane ^(H)^	2100	2100	4.4	-
62	*α*-Linolenic acid ^(F)^	2100	2102	1.4	-
63	Phytol ^(T)^	2104	2104	1.1	-
64	Octadecyl acetate ^(E)^	2160	2161	-	2.0
65	Isoangenomalin ^(C)^	2182	2186	0.5	-
66	(*Z*)-2-(9-Octadecenyloxy)-ethanol ^(Ac)^	2336	2336	0.4	-
67	Behenic alcohol ^(Ac)^	2470	2470	0.8	-
68	Heptacosane ^(H)^	2700	2700	2.3	-
69	Octacosane ^(H)^	2800	2800	3.6	-
70	3*α*-24-Propylidene-cholest-5-en-3-ol ^(S)^	2881	2880	-	3.4
71	*α*-Sitosterol ^(S)^	3065	3066	0.8	5.2
72	*α*-amyrin ^(T)^	3322	3320	0.6	-
73	Tetratetracontane ^(H)^	4392	4395	4.7	-
	Hydrocarbons (Sr. No. 2, 3, 10, 11, 20, 25, 28, 32, 40, 56, 58, 61, 68, 69, 73)			24.1	8.0
	Benzenes (Sr. No. 6, 9, 12, 14, 16, 18, 19, 23, 24, 54)			7.5	18.8
	Alcohols (Sr. No. 4, 5, 8, 15, 52, 66, 67)			34.1	3.5
	Aldehydes (Sr. No. 1, 7, 17)			2.1	1.2
	Esters (Sr. No. 46, 59, 60, 64)			1.9	5.7
	Ketones (Sr. No. 34, 35)			3.5	13.9
	Phenolics (Sr. No. 26, 38, 51)			1.6	8.7
	Fatty acids (Sr. No. 30, 41, 57, 62)			4.9	8.8
	Terpenoids (Sr. No. 13, 21, 22, 27, 29, 31, 33, 36, 37, 39, 42–45, 47–50, 53, 55, 63, 72)			16.2	17.0
	Steroids (Sr. No. 70, 71)			0.8	8.6
	Coumarins (Sr. No. 65)			0.5	-
	Total			97.2	94.2

KRI: Kováts retention index; LRI: Linear retention index; RA%: Relative amount; (-) = Absent; Ac: Alcohols, Ad: Aldehydes, B: Benzenes, C: Coumarins, E: Esters, F: Fatty acids, H: Hydrocarbons, K: Ketones, P: Phenolics, S: Steroids, T: Terpenoids; Sr. No.: Serial number of compounds.

**Table 2 molecules-28-02774-t002:** Antioxidant capacities of leaf and stem oils in *F. japonicum*.

Sample	IC_50_ (mg/mL, *n* = 3)	
	DPPH	·OH
Leaf oil	20.56 ± 0.20	1.52 ± 0.03
Stem oil	9.22 ± 0.11	0.90 ± 0.02
Vitamin C ^a^	4.10 ± 0.18	0.65 ± 0.01

Notes: DPPH, 1,1-diphenyl-2-pinylhydrazyl; ·OH, hydroxyl free radical. ^a^ Used as a positive control.

**Table 3 molecules-28-02774-t003:** MIC values of leaf and stem oils of *F. japonicum*.

Strains of Bacteria	MIC (mg/mL)	
	Leaf Oil	Stem Oil
Methicillin-sensitive *Staphylococcus aureus*	0.62	0.16
Methicillin-resistant *Staphylococcus aureus*	0.31	0.16
*Pseudomonas aeruginosa*	0.62	0.31
*Escherichia coli*	1.25	0.62
*Proteus* spp.	0.62	0.31
*Klebsiella pneumoniae*	1.25	0.62

## Data Availability

Not applicable.

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
