# Peer review of "Composition and Antioxidative and Antibacterial Activities of the Essential Oil from Farfugium japonicum"

_molecules, 2023, doi:10.3390/molecules28062774_

Round 1

Reviewer 1 Report

The manuscript "Composition, Antioxidative and Antibacterial Activities of Essential Oil from Composition, Antioxidative and Antibacterial Activities of Essential Oil from Farfugium japonicum" is interesting and has good scientific merits but some points required clarification by the authors.

among these points:

1- The yield of the oils should be mentioned in term of % w/w or %v/w otherwise I think with the low amount of phenolic components it will be hard to accept that IC50 in DPPH will be n terms of microgram and not milligram

2- On what basis the authors selected the antioxidant and antibacterial activity ?

3- The chromatograms should be given as supplementary 

4- I would rather recommend addition of a column in the identification table about the reported KI from Adams 2007"Identification of Essential Oil Components by Gas Chromatography/ quadrupole Mass Spectroscopy"

5- More discussion is required to explain the differences between the leaves and stems essential oils from biosynthesis point of view 

6- The manuscript should be checked by an English native speaker to remove some syntax and typos 

Author Response

1- The yield of the oils should be mentioned in term of % w/w or %v/w otherwise I think with the low amount of phenolic components it will be hard to accept that IC50 in DPPH will be n terms of microgram and not milligram

Answer: After checking the data, we found that the calculation error of volatile oil percentage of 0.11% and 0.06%, and 1.1 %, 0.6 % v/w are correct. The measurement unit was wrong and IC50 in DPPH should be milligram. They were corrected.

2- On what basis the authors selected the antioxidant and antibacterial activity ?

Answer: Because the antioxidant and antibacterial activities of volatile oil are relatively extensive in the literature, but there may be differences due to different stems and leaves of plants, which is the main basis for less reports

3- The chromatograms should be given as supplementary 

Answer: The chromatograms are seen in Figure1.

4- I would rather recommend addition of a column in the identification table about the reported KI from Adams 2007"Identification of Essential Oil Components by Gas Chromatography/ quadrupole Mass Spectroscopy"

Answer: I am sorry to try my best to order the free e-book of "Identification of Essential Oil Components by Gas Chromatography/ quadrupole Mass Spectroscopy" (https://diabloanalytical.com/ms-software/essentialoilcomponentsbygcms/request-for-e-book-essential-oil-components-by-gc-ms/), but I don't receive the reply in my email box. If the reviewer can provide it, I can correct it.

5- More discussion is required to explain the differences between the leaves and stems essential oils from biosynthesis point of view 

Answer: Taken terpenoids as an example, the biosynthesized procedure and relative enzymes, genes were introduced to explain the difference between the leaves and stems essential oils, as seen in Figure 3 and relative introduction.

6- The manuscript should be checked by an English native speaker to remove some syntax and typos 

Answer: It was checked by MDPI English Editing Service.

Reviewer 2 Report

Dear Editor/Author;

-          Abstract should be detailed.

-          The introduction should be expanded extensively.

-          Emphasis should be placed on the purpose of the article.

-          The article should be reviewed by a native English reader.

-          Grammatical errors should be corrected throughout the article.

-          The conclusion should be strengthened.

-          The contribution of this article to future studies should be stated.

-          It should be clearly stated in which area this article fills the gap.

Author Response

 -Abstract should be detailed.

    Answer: It was corrected.

-The introduction should be expanded extensively.

    Answer: It was corrected.

- Emphasis should be placed on the purpose of the article.

    Answer: It was corrected.

- The article should be reviewed by a native English reader.

    Answer: It was checked by MDPI English Editing Service.

- Grammatical errors should be corrected throughout the article.

    Answer: It was checked by MDPI English Editing Service.

- The conclusion should be strengthened.

    Answer: It was corrected.

- The contribution of this article to future studies should be stated.

    Answer: It was corrected.

- It should be clearly stated in which area this article fills the gap.

    Answer: It was corrected in the conclusion.

Reviewer 3 Report

1. Language editing by a native speaker or a professional English editor will significantly improve the reader’s appreciation of the work presented in the manuscript.

2. Genus and species name of the reported plant need to be in italics (Line 63).

3. Lines 14 and 202 should be paraphrased.

4. Acronyms should be written in full for the first time (see line 17)

5. Line 71 dominant was written with a capital D; is there any reason for this?

6. There should be a space between a number and unit. See lines 70-78. Please, correct throughout the manuscript!

7. Line 196, 1000 g should be written in words since it is starting the sentence.

8. The manuscript should be accepted after making these corrections.

Author Response

      1. Language editing by a native speaker or a professional English editor will significantly improve the reader’s appreciation of the work presented in the manuscript.

Answer: It was corrected.

  1. Genus and species name of the reported plant need to be in italics (Line 63).

Answer: It was corrected.

  1. Lines 14 and 202 should be paraphrased.

Answer: It was corrected.

  1. Acronyms should be written in full for the first time (see line 17)

Answer: It was corrected.

  1. Line 71 dominant was written with a capital D; is there any reason for this?

Answer: Sorry, it is a mistake and has been corrected.

  1. There should be a space between a number and unit. See lines 70-78. Please, correct throughout the manuscript!

Answer: It was corrected.

  1. Line 196, 1000 g should be written in words since it is starting the sentence.

Answer: It was corrected.

Reviewer 4 Report

The article Composition, Antioxidative and Antibacterial Activities of Essential Oil from Farfugium japonicum, characterize volatile compounds of leaf and stem form F. japonicum and the antioxidant and antibacterial capacities of the herb after supercritical fluid extraction of CO2 process.

The work is well structured; however the Hydroxyl Radical Inhibition Test should be described in more details, since the classical deoxyribose method was not applied, but a modification of Wang et al. method [29], which is available only as abstract. In order to assess the Hydroxyl radical (.OH) Inhibition test changes in the analytical procedure some validation parameters of the method should be given as well.

Author Response

The work is well structured; however the Hydroxyl Radical Inhibition Test should be described in more details, since the classical deoxyribose method was not applied, but a modification of Wang et al. method [29], which is available only as abstract. In order to assess the Hydroxyl radical (.OH) Inhibition test changes in the analytical procedure some validation parameters of the method should be given as well.

Answer: It was corrected. pleas see the attachment. Thanks a lot.

Round 2

Reviewer 1 Report

The authors responded positively with all the raised points

It could be accepted n the present form 

Author Response

Thank you for the reviewer's suggestion and decision.
